# The Impact of Export Promotion Programs on Export Competitiveness and Export Performance of Craft Products

Saksuriya Traiyarach [ID] and Jantima Banjongprasert *

Silpakorn University International College, Silpakorn University, Charoen Krung Road, Bangkok 10500, Thailand; traiyarach_s@silpakorn.edu
* Correspondence: banjongprasert_j@silpakorn.edu

**Abstract:** International trade is defined as economic transactions between countries worldwide. Promoting the export of craft products, which are valued products, is critical for international business as the sales growth increases worldwide. Moreover, the export of craft products has increased international trade and maximized economic value in the highly competitive global market. Therefore, businesses need to be promoted to increase their competitiveness. This study explores the impact of export promotion programs on export competitiveness and the performance of craft products. A self-administered questionnaire was used to correct the data. There were 400 respondents completing the questionnaires, who were working in craft product export companies using marine transport. The data analysis is conducted by using Structure Equation Modelling (SEM). The findings show that the export promotion program has a significant positive relationship with export competitiveness. A positive relationship between export competitiveness and export performance is also found. The results indicate that export competitiveness fully mediates the relationship between export promotion programs and export performance. The findings from this study contribute to craft product export businesses and provide a practical exporting approach. Marine transport is one of the critical international entry modes many companies use to expand businesses. It should be noted that shipping cost savings are related to export efficiency.

**Keywords:** export promotion programs; export competitiveness; export performance; marine transportation; craft products

## 1. Introduction

Recently, it has been impossible to avoid the topic of globalization and the global market, in which enterprises compete against foreign rivals in their home markets, even if they have not yet internationalized themselves [1]. Consequently, in a more dynamic and competitive global marketplace, businesses are pushed to expand beyond their national boundaries and boost their competitiveness in both local and international markets and within their own industries. At this stage, exporting becomes a tool for businesses to use in order to extend their reach into new international markets, increase their sales, and more effectively compete against global competitive pressures [2].

Export business has significant impacts on investment expansion and creating labor demand at both national and international levels [2,3]. Many enterprises need to establish their branches in other countries to ensure that they have a place that can store their products when they are exported or they need to tie alliances with national and international partners to facilitate the conveniences of exportation activities [4]. This expansion requires substantial financial capabilities and a large investment in resources. In addition, the export business can assist in the import of foreign currency as well as contribute to the efficient use of resources because entrepreneurs have to consider effectively utilizing and flowing business resources such as labor, technologies, methods, materials, and finances, on a national and international level in order to obtain business competitiveness [5]. Lastly, the

export business can create added value to resources because the entrepreneurs are required to create and provide product and service exportation to satisfy the different customers' preferences. Herewith, the entrepreneurs must attempt to understand the customers' needs and demands from different perspectives, including cultures, technologies, politics, and economic relevancies [6].

However, export entrepreneurs are encountering difficulties in achieving export performance. For example, the exporters lack professional, proactive marketing skills as well as export-related knowledge such as technology provision, export business know-how, and international experiences to compete with foreign competitors. In addition, the problem of high labor costs, which affects production costs, causes the prices of products and services to rise, which in turn increases the inability to compete with other competitors that can gain competitive advantages [7,8]. Furthermore, Jung et al. [9] reveal that the important analysis of decision-making factors for selecting an international freight transportation mode can influence business performance. At the same time, Fulzele et al. [10] suggest that it is necessary to consider the selection of transportation modes that can be linked to business success and sustainability.

Marine transport is one of the major forms of exportation. Given its importance, marine transport has been extensively studied in previous research. Many publications on maritime transport-related topics have been published in the last few decades [11]. Marine transport comprises a large variety of operations and, in combination with port activities and logistic hubs, has a substantial impact on the development of marine industry and trade, hence promoting economic growth and employment creation [12]. In Thailand, marine transportation is crucial for the country since there are various areas with access to rivers, seas, and oceans that are trading products with other counties such as China, Hong Kong, and so on [13,14]. Krailasuwan [13] indicated that maritime transportation accounted for about 90% of Thai imports and exports. In order to determine how to advance maritime trade liberalization, the CHINA–ASEAN free trade area regionally liberalizes marine service trade to facilitate international transportation in global trade [15]. However, Wei [14] and Lu [15] report that having international trade to export and import products to and from foreign countries, such as Thailand, requires considering ways to improve competitive business advantages such as satisfactory prices, product uniqueness as well as government support.

Freixenet and Churakova [16] found that export promotion programs can assist entrepreneurs in reaching their goals in creating firms' export competencies. Additionally, Coudounaris [17] demonstrates that the export promotion program can affect export performance. Therefore, it is necessary to study how to improve the export performance for the export business by studying export promotion programs that can bring about better business competitiveness and performance. This study aims to study the impact of export promotion programs on export competitiveness and export performance in the craft product export firms in Thailand since these firms have promoted craft products made by people from the community. If these firms can gain export competitiveness and export performance, they can finally create sustainable community development and links to increase the national economy. In addition, this article aims to suggest the implication of using marine transport. To display the study content, this article begins with an introduction that describes the significance of this study and the objectives. The next section portrays the study materials and methodology. Then, the research findings as well as a discussion of the results, research contribution, and future research recommendations, are composed.

## 2. Materials and Methods

### 2.1. Export Promotion Program and Export Competitiveness

Export competitiveness is significant for the export business because it can positively influence export performance, including financial and market performance [18,19], a nation's sustainable development and foreign trade [20], and advantages in emerging markets [21]. Various studies [17,22–25] mention that the export promotion program pro-

vided by the government increases export competitiveness. This refers to the role of the government in offering the support related to the export, providing activities encompassing financial and tax incentives, trade fairs participation, market research (specific information), export training, and export consultation [26]. Similarly, Coudounaris [17] defines the export promotion program as the function of the governments or associated governmental entities that aim to encourage the exporting activity of a nation via its local businesses. Indeed, export promotion activity can support multi-modal freight transportation since it can improve the sustainability of the business by minimizing the transportation expenses, product damages, released pollutants, and road congestion, as well as by boosting the delivery speed. In addition to adopting export promotion activities for marine transport, Akims and Danyil [27] reveal that infrastructures such as electronic export clearance and the stability of the country's power supply via the ongoing expansion of energy distribution lines could lead to benefit export performance and finally link to export-led economic growth.

Consequently, the terms of the export promotion program found various advantages to all levels, including international, national, sectoral, and company levels from public, public or multi-sectoral non-profit, non-profit, and private sectors [28]. The export promotion program can influence firms' competitiveness, including product differentiation, product quality, and promotion effectiveness [22–25]. In addition, Freixanet and Churakova [16] show that export promotion programs are important for Russian manufacturers since they can assist the entrepreneurs in reaching their goals in creating firms' export marketing competencies by highlighting clear issues regarding program awareness, availability, and accessibility related to exportation. According to Njinyah [29], government policies for export promotion may provide possibilities for businesses to acquire competitiveness in terms of nation and market characteristics, understanding of trade obstacles in the export market, and the correct product marketing mix. Similarly, Catanzaro and Teyssier [30] revealed that export promotion programs might be associated with the growth of export capabilities and the adoption of risk management techniques. According to Mata et al. [31], financial and marketing support from government policy may help young entrepreneurial firms to overcome the susceptibility of being new and small while gaining competitive export capabilities. Adedoyin et al. [32] added that the government should ensure macroeconomic and political stability to achieve export. Accordingly, we propose the hypothesis that H1: export promotion program can influence export competitiveness.

### 2.2. Export Competitiveness and Export Performance

Export competitiveness refers to the ability to take action on exportation compared to other competitors, such as product differentiation, product quality, and promotion effectiveness [22–25]. In terms of product differentiation, Guru and Paulssen [25] and Priede [33] explain that entrepreneurs can have the ability to produce unique products for the export market, have a differentiated image compared to competitors in the international market, and create a design that is in accordance with the wishes of consumers. Meanwhile, Boehe, and Barin Cruz [22] and Guru and Paulssen [22] demonstrate that entrepreneurs can have export competitiveness in terms of product quality by having products that meet a standard and that are superior compared to the quality of their competitors in the international market. Similarly, Freixanet [23] illustrates that exportation entrepreneurs can gain export competitiveness by obtaining the direct export promotion, receiving information on markets, programs, or export know-how, and use of foreign trade offices, having consultancy, seeking investment support, obtaining sales leads as well as improving marketing managers' international orientation. Export competitiveness has a positive effect on export performance. As regards export performance, it can be categorized into two main dimensions, including financial and market performance [18,19]. The financial performance involves profitable foreign operations, strong sales, quick expansion, and enhanced profitability. Market performance refers to the international operations that enhance international competitiveness, strengthen the firms' strategic position, significantly increase the international market share, establish a foothold in new markets, identify export

opportunities, and increase product awareness in export markets. Various studies indicate the effect of export competitiveness on export performance. For instance, Boehe and Barin Cruz [22] found that export competitiveness in terms of product differentiation can be linked to export performance. In the meantime, Priede and Pereira [24] show that EU promotion program competitiveness can establish export performance. Lastly, Keskin et al. [34] explain that competitiveness, including differentiation, can increase export companies to obtain export performance in foreign markets. Accordingly, the hypothesis can be that H2: export competitiveness can influence export performance.

### 2.3. Export Promotion Program and Export Performance

Various studies indicate the effect of export promotion programs on export performance since the support from the government and related export agencies such as financial and tax incentives, trade fairs participation, market research (specific information), export training, and export consultation can enhance the entrepreneurs gain export performance from both financial and market perspectives such as profitability, high volume of sales, export opportunities, international experiences, a firm's strategic position as well as a foothold in the new market [18,19,28]. Moreover, Coudounaris [17], studying export promotion programs assisting SMEs, demonstrates that the export promotion program can influence the export performance, including knowledge, capabilities, competitive advantage, and business experience. Accordingly, the hypothesis can be that H3: export promotion program can influence export performance.

### 2.4. Export Promotion Program, Export Competitiveness and Export Performance

There are studies regarding the significant impact of export promotion programs on export competitiveness and export performance. For example, Njinyah [29] found that government policies for export promotion can be linked to creating competitiveness in terms of exportation. Additionally, Catanzaro and Teyssier [30] reveal that export promotion programs can assist the entrepreneurs in obtaining the growth of export capabilities. In addition, Coudounaris [17] demonstrates that the export promotion program could link the export performance, including knowledge, capabilities, competitive advantage, and experience. Meanwhile, a study by Keskin et al. [34] also revealed that unique company capabilities could provide the opportunity for the export businesses in international markets and increase the performance of foreign markets' exports. However, there are some studies demonstrating that export promotion programs may not directly influence export performance. For example, Njinyah [29] found that the government's export policy did not directly affect export performance. Moreover, Mata et al. [31] indicate that the support from related institutions requires competitive capabilities to play a mediating role, which can finally affect the enterprises' export performance. Accordingly, the hypothesis can be that H4: export competitiveness mediates the relationship between export promotion program and export performance. All hypotheses are illustrated in Figure 1.

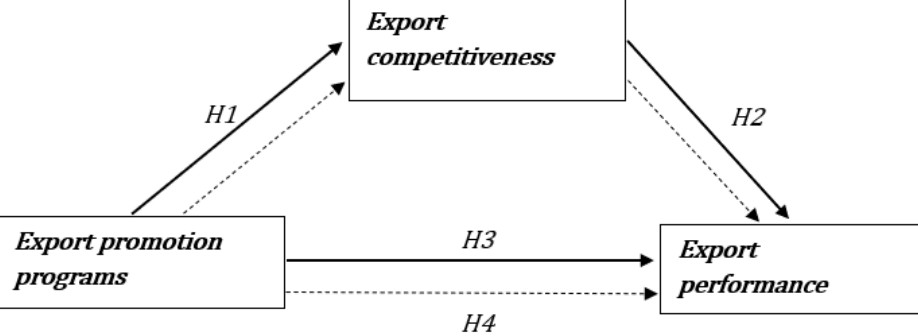

**Figure 1.** Conceptual Framework.

*2.5. Research Methods*

In response to the study objectives, the study was then designed to adopt a quantitative research approach. The population in this study was craft product companies in Thailand. The sample size was calculated according to the general rules of choosing the Alpha level of confidence, and the acceptable error value is Alpha = 0.05 and acceptable error values of 5%, which are considered suitable values Krejcie and Mogan [35] because in this study the population numbers were unknown; therefore, the proportion of the population that needs to be chosen is set at 20% or 0.2 at the confidence level of 95%, and the error value of 5% or 0.05, by using the formula to calculate the suitable number of samples by using the methodology of Cochran's method [36]. This method is appropriate for calculating the sample size of a large population whose degree of variability is unknown [37,38]. As many craft companies are exporting their products abroad, the number of companies and their employees is unknown. Furthermore, there is a lack of information regarding the product types and the size of businesses. Thus, Cochran's approach [36] was used to determine the sample size of this study. The formula and calculation are presented below.

The Cochran formula is:

$$n_0 = \frac{Z^2 pq}{e^2}$$

where:
$e$ is the desired level of precision (i.e., the margin of error),
$p$ is the (estimated) proportion of the population that has the attribute in question,
$q$ is $1 - p$.

Replaced by the following.

The study set $p$ equal to 0.5, and 95% confidence was the target, allowing at least $\pm 5$ percent error. Herewith, a 95% confidence level provides $Z$ values of 1.96. Accordingly, the corresponding numbers are assigned as follows.

$$((1.96)2x\,(0.5)\,(0.5))/(0.05)2 = 385.$$

Consequently, there were 400 (rounded) employees from craft product companies using marine transport that were selected at a confidence level of 95%, which covered the appropriate sample according to Cochran's formula [36], while also obtaining larger samples. This was an advantage in terms of reliability and representation based on the population of Newman [39]. The research tool was a questionnaire survey. Prior to data collection, a systematic item-objective congruence (IOC) with five experts from marketing, international business, and statistic fields was used to indicate content validity. Cronbach's alpha drawn from 50 sets of pretests was employed to analyze the item reliability. Regarding the pretest study, the result indicated that most of the respondents were female (50.0%), aged between 31–40 years old (46.0%), graduated with lower than a bachelor's degree (58.0%), worked as operational staff, and had less than 5 years of export experience. In addition, the analysis indicated that the IOC is equal to 0.942 and Cronbach's alpha is 0.980, indicating that the research instrument has the appropriate quality [40].

In terms of the measures in this study, there are three major perspectives: export promotion programs, export competitiveness, and export performance. Export promotion program items were adopted from Cuyvers et al. [28] using a 7-rating scale, where 1 signifies extremely unimportant and 7 signifies extremely important. For export competitiveness, there are three dimensions: product quality, product differentiation, and promotion effectiveness. Firstly, for product quality, the items were adopted from Boehe and Barin Cruz [22], Guru and Paulssen [25], and Magnani and Zucchella [41] meanwhile, the items associated with product differentiation were adopted from Boehe and Barin Cruz [22], Guru, and Paulssen [25], and Priede [33]. Lastly, the items pertaining to promotion effectiveness were adopted from Freixanet [23] using a 7-rating scale indicating 1 to refer to extremely unimportant and 7 to refer to extremely important. For export performance, there are two concepts: financial performance and market performance adopted from



Francis and Collins-Dodd [18] and Solberg and Durrieu [19], asking about profitability, sales volume, operation growth, market share increase, export opportunity, export awareness, and strategic market position using 1 to refer extremely unsatisfactory and 7 to refer extremely satisfactory.

To obtain the data, the questionnaire was distributed to 400 employees from 46 exporting craft product companies using marine transport by employing the purposive sampling method. The data were collected from February to April 2022. Cronbach's alpha was used to indicate the data reliability. From the study, Cronbach's alpha was 0.921 for the export promotion program, 0.969 for export competitiveness, and 0.936 for export performance.

Then, confirmatory factor analysis (CFA) was used to assess the model fitness as well as convergent and discriminant validity, as shown by factor loading (FL > 0.5), composite reliability (CR > 0.5), average variance extracted (AVE > 0.5), correlation matrix, and the square root of AVE. It was predicted that the examined model would give satisfactory goodness-of-fit indices which include Chi-square Probability Level (*p*-value > 0.05), Relative Chi-square (CMIN/df < 3), Goodness of Fit Index (GFI > 0.90), Root Mean Square Error of Approximation (RMSEA < 0.08), Root mean square residual (RMR < 0.05), Tucker and Lewis (TLI > 0.90), Normed Fit Index (NFI > 0.90), and Adjusted Goodness of Fit (AGFI > 0.90). However, when the model was determined to be unfit, it was permitted to be altered using modification indices. For hypothesis testing, structural equation modeling (SEM) with bootstrapping technique was employed [40]. After the result of the study was drawn, the findings were explained and discussed.

### 3. Results

#### 3.1. Respondents' Profiles

From 400 employees, the majority of the workers who responded to the surveys were female (47.0%), male (46.0%), and others (7.0%). Regarding the respondents' age, there were 41.3 percent aged between 31 and 40 years old, 30.3 percent aged between 21 and 30 40 years old, 17.3 percent aged between 41 and 50 40 years old, 4.5 percent aged below 20 40 years old, and 0.8 percent aged older than 60 40 years old. In terms of education level, the majority of respondents graduated with less than a bachelor's degree (47.0%), followed by a bachelor's degree (38.5%), and above a bachelor's degree (14.5%). In terms of their position, 59.8 percent are in operational roles, while 40.3% are in management roles such as firm owners, executives, department managers, and division managers. Lastly, 46.8 percent had less than 10 years of experience working in exporting organizations. Meanwhile, 38.8 percent had 10–20 years of experience, and 14.5 percent have more than 20 years of experience.

#### 3.2. Model Development, Convergent Validity, and Discriminant Validity

Confirmatory factor analysis was conducted to investigate convergent validity and discriminant validity of export promotion programs, export competitiveness, and export performance. Good-fitness indices: $\chi^2/\mathrm{df} \leq 3.00$, GFI $\geq 0.90$, CFI $\geq 0.90$, NFI $\geq 0.90$, AGFI $\geq 0.90$, RMSEA $\leq 0.07$, and RMR $\leq 0.08$ were considered both before and after model adjustment. The initial model revealed unacceptable values: Cmin/df of 6138, *p*-value of 0.000, GFI of 0.830, AGFI of 0.765, RMR of 0.042, RMRSEA of 0.113, TLI of 0.907, CFI of 0.923, and NFI of 0.910. However, the value of good-fitness indices was better when the adjusted model was tested. These values included: Cmin/df of 1.054, *p*-value of 0.366, GFI of 0.981, AGFI of 0.959, RMR of 0.017, RMRSEA of 0.012, TLI of 0.999, CFI of 0.999, and NFI of 0.990. Furthermore, the factor loadings, composite reliability, and average variance extracted from the analyzed variables were taken into account to explain unidimensional measurements; they were more than 0.05. [40]. All of the variables' related values are shown in Table 1.

**Table 1.** Factor Loading and convergent validity.

| Variables | EXPPR | EXCOM | EXPER | CR | AVE |
|-----------|-------|-------|-------|-----|-----|
| EXPPR1 | 0.733 | | | | |
| EXPPR2 | 0.867 | | | | |
| EXPPR3 | 0.872 | | | 0.983 | 0.836 |
| EXPPR4 | 0.893 | | | | |
| EXPPR5 | 0.816 | | | | |
| EXCOM1 | | 0.912 | | | |
| EXCOM2 | | 0.932 | | 0.948 | 0.895 |
| EXCOM3 | | 0.840 | | | |
| EXPER1 | | | 0.838 | | |
| EXPER2 | | | 0.916 | | |
| EXPER3 | | | 0.861 | | |
| EXPER4 | | | 0.846 | 0.958 | 0.850 |
| EXPER5 | | | 0.810 | | |
| EXPER6 | | | 0.901 | | |
| EXPER7 | | | 0.775 | | |

Note: EXPPR1–5 refers to export promotion program, EXCOM1–3 refers to export competitiveness, and EXPER1–7 refers to export performance.

From Table 1, the study revealed that all factor loadings from CFA were about 0.733–0.893 for the export promotion program, 0.840–0.932 for export competitiveness, and 0.775–0.916 for export performance. Meanwhile, the result also revealed composite reliability ranging from 0.948–0.983 and average variance extracted from 0.836–0.895. These values were more than 0.50, indicating that all variables can be studied further [40]. In addition, the correlation matrix and square root of AVE were considered for convergent validity and discriminant validity. Table 2 shows discriminant validity.

**Table 2.** Discriminant validity.

| Variables | EXPPR | EXCOM | EXPER |
|-----------|-------|-------|-------|
| EXPPR | 0.914 | | |
| EXCOM | 0.899 | 0.946 | |
| EXPER | 0.756 | 0.857 | 0.922 |

Note: EXPPR refers to export promotion program, EXCOM refers to export competitiveness, and EXPER refers to export performance.

From Table 2, the square root of AVE was higher than the correlation coefficient matrix of the variables, which correlation coefficients ranging from 0.756–0.899. In addition, the variance inflation factor (VIF) to inspect multicollinearity ranged from 2.179–4.631. This means that all variables, including export promotion program, export competitiveness, and export performance were identical and appropriate for further analysis.

### 3.3. Finalized Model and Hypothesis Analysis

After assessing convergent and discriminant validity using confirmatory factor analysis (CFA), the final model was created, and structural equation modeling (SEM) was used to test the hypothesis. Consequently, the final model was initially investigated, and its good-fitness indices were unacceptable since they did not meet the recommended criteria: Cmin/df of 6.138, *p*-value of 0.000, GFI of 0.830, AGFI of 0.765, RMR of 0.042, RMRSEA of 0.113, TLI of 0.907, CFI of 0.923, and NFI of 0.910. However, the model was modified based on the proposal of modification indices, and its good-fit indices were subsequently enhanced, Cmin/df of 0.874, *p*-value of 0.735, GFI of 0.984, AGFI of 0.966, RMR of 0.013, RMRSEA of 0.000, TLI of 1.000, CFI of 1.000, and NFI of 0.992 (see Figure 2).

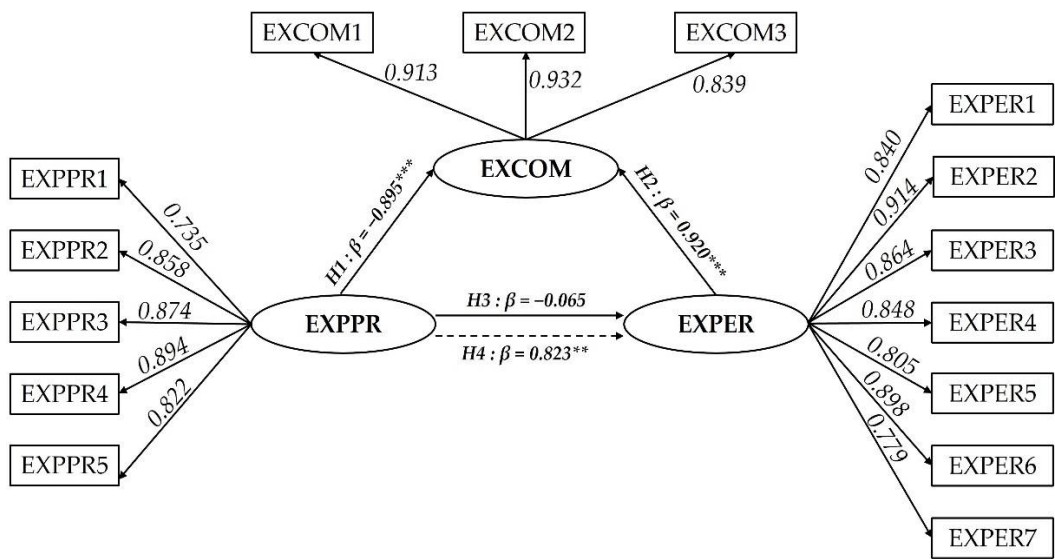

**Figure 2.** Finalized Model.

Table 3 shows the findings of the hypothesis investigation and the influence prediction ability on variables. Hypothesis (H) 1 showed that export promotion program influenced export competitiveness ($\beta = 0.895$, $p < 0.05$) while hypothesis (H) 2 implied that export competitiveness influenced export performance ($\beta = 0.920$, $p < 0.05$). Nevertheless, hypothesis (H) 3 revealed that the export promotion program did not influence export performance ($\beta = -0.069$, $p < 0.05$) at the statistically significant level of 0.001.

**Table 3.** Standardized Estimate, Unstandardized Estimate, Standard Error, *t*-value, *z*-value, and *p*-value.

| Variables | Untandardized Estimate (b) | Standardized Estimate (β) | S.E. | $t-$Value | $p-$Value |
|---|---|---|---|---|---|
| H1: EXPPR → EXCOM | 0.928 | 0.895 | 0.050 | 18.412 | *** |
| H2: EXCOM → EXPER | 0.746 | 0.920 | 0.074 | 10.036 | *** |
| H3: EXPPR → EXPER | −0.055 | −0.065 | 0.069 | −0.800 | 0.424 |

Note: EXPPR refers to export promotion program, EXCOM refers to export competitiveness, and EXPER refers to export performance. *** *p*-value =< 0.001.

Table 4 indicates that the export promotion program in hypothesis (H) 4 had a significant indirect influence with a fully mediating role on export performance through export competitiveness since the *p*-value from bootstrapping technique was lower than 0.05.

**Table 4.** Mediating Effect Result.

| Variables | IV-N-DV | | Mediating Type |
|---|---|---|---|
| | Direct | Indirect | |
| H4: EXPPR → EXCOM → EXPER | −0.065 | 0.823 ** | Full Mediation |

Note: EXPPR refers to export promotion program, EXCOM refers to export competitiveness, and EXPER refers to export performance. ** *p*-value =< 0.01.

## 4. Discussion

The findings of the study reveal that export promotion programs influenced export competitiveness. This is because the government and related agency support programs such as financial and tax incentives, participation in trade fairs, export-related market research, export training, and export support and consulting can enable export entrepreneurs

to obtain export competitiveness, including high product quality differentiation, and promotion effectiveness. They may use the government-provided consultancy, market information, and export training program to connect and develop the skills and abilities of their personnel so that they can successfully use organizational resources for their export efforts. This is consistent with the research conducted by Cuyvers et al. [28], who examined a decision support model for the planning and evaluation of export promotion activities by government export promotion institutions in the Belgian case and found that financial and tax incentives, trade fairs, market information, export training, and education, and export consultancy can help evaluate export promotion activities. Meanwhile, Freixanet and Churakova [16] express that export promotion programs in Russian manufacturers successfully achieve their objectives of enhancing firms' export marketing competencies by highlighting apparent issues regarding program awareness, availability, and accessibility-related to exportation. Export promotion policies in Cameroon were studied by Njinyah [29], who found that government policies for export promotion can create opportunities for entrepreneurs to gain competitiveness in terms of country and market specifications, knowledge of trade barriers in the export market, as well as the right marketing mix of the product. Finally, Catanzaro and Teyssier [30] indicate that export promotion programs might be linked to the development of export capabilities as well as the adoption of risk management methods. The study is also in line with the result of Mata et al. [31], who indicated that finance and marketing assistance might aid young entrepreneurial enterprises in overcoming the vulnerability of newness and smallness in acquiring competitive export capacities. To conclude with a discussion, the export promotion programs have a significant impact on export competitiveness.

In addition, the study result indicates that export competitiveness, including product quality, product differentiation, and promotion effectiveness influenced export performance regarding financial and market performance. This is due to the fact that the entrepreneurs' competitiveness in producing high-quality products that meet international standards and customer requirements as well as in offering product differentiation that specifically meets the unique customer demands and differs from competitors can be linked to export performance such as international operation improvement, profitability, sales volume growth, and international opportunity. According to the study, entrepreneurs' competitiveness in gaining promotion effectiveness—receiving direct promotion, exportation consulting, investment support, and enhancing marketing managers' international orientation—can also improve export performance from both a financial and market perspective. The study is consistent with the study conducted by Catanzaro and Teyssier [30], who demonstrate that export capabilities in managing international risk and implementing foreign direct investment strategies could better the international performance of small and medium enterprises. Furthermore, the findings of this research are in agreement with those of Keskin et al. [34], who indicated that unique company capabilities, comprising informational, relational, and marketing skills as well as differentiation and cost leadership, can offer export businesses a competitive edge in international markets and increase their export performance in foreign markets simultaneously. Accordingly, export competitiveness can be confirmed to have an impact on export performance.

Unfortunately, the findings revealed that the export promotion program did not influence export performance. However, the bootstrapping technique indicated that the export promotion program had a significant indirect influence on export performance. This is due to the fact that export promotion programs such as financial and tax incentives, participation in trade fairs, export-related market research, export training, and export support and consulting provided by the government and related export agencies are not enough to directly create export performance regarding international operation improvement, profitability, sales volume growth, and international opportunity. To link the export promotion programs to export performance, the entrepreneurs should be able to manage themselves to gain export competitiveness in creating high product quality, product differentiation, and promotion effectiveness. For this result, the study corresponds to the study carried out

by Njinyah [29], who found the export policy from the government did not have a direct effect on export performance from both financial and non-financial perspectives, including the increased scale of business, return on investment, profit growth, and relationship with the stakeholder. In addition, the study is also consistent with Mata et al. [31], indicating that competitive capabilities play a significant role in mediating the relationship between marketing support from related institutions and the export performance of the enterprises. Lastly, the study is supported by Catanzaro and Teyssier [30], who also implied that export capabilities play a crucial role in the mediation between export promotion programs offered by the government and the international performance of the small and medium businesses.

The result of this study can be linked to have a managerial, theoretical, and political contribution. For managerial contribution, to create a satisfactory export performance for craft products exportation business, the entrepreneurs should be determined to create export competitiveness, including product quality, product differentiation, and promotion effectiveness by attempting to provide unique products differing from competitors with high product quality through utilizing direct promotion, information, consultancy, investment support, marketing managers' international orientation related to exportation. However, to create competitiveness among export enterprises, the entrepreneurs need support from the government, as depicted in the study result indicating that the export promotion programs provided by the government and relevant export agencies play a significant role in bettering export performance via export competitiveness. For the theoretical contribution, the study result can respond to the literature reviews to confirm that the export promotion program had a significant indirect influence with a fully mediating role on export performance through export competitiveness. For the political contribution, the government and relevant export agencies should emphasize providing the export promotion programs such as financial and tax incentives, domestic and international trade fairs participation, export-related market information, export training, and education, as well as export consultation. Since marine transport plays a significant role in the exportation of many countries, the government should also establish appropriate infrastructures such as electronic export clearance, the country's power supply, and others to benefit export performance.

Nevertheless, this study contains numerous limitations that can be linked to finding future potential research. Firstly, the study only focuses on examining the effect of export promotion programs on export competitiveness and export performance. There are numbers of related influential factors such as differentiation and niche strategy, alliances with international partners, or distribution effectiveness that are ignored. Therefore, future research should extend the study to cover other potential factors. Secondly, this study obtained data for analysis from only a single industry that is related to craft product exportation firms. Consequently, the future study can shift to other business sectors such as agricultural, industrial, electronic, or food and beverage products. Some of these are encountering export problems. Meanwhile, some of these have the potential to develop the national economy. Thirdly, the study aimed at revealing export competitiveness and export performance in general, ignoring the study on significant characteristics and barriers regarding the destination of the importers' country. Some literature reviews write that different importer countries require different requirements varying export competitiveness and export performance. Future research should perhaps be conducted on specified countries. Another point is that this study focused on overall transportation, which lacks the study of specific mode, especially the marine mode. Since marine transportation bears a significance to entrepreneurs and nations, future research should specify factors related to marine transportation to create export performance effectively. Lastly, this study is devoted to research analysis through a quantitative research approach. So, the other research approach, such as qualitative research orientation with different data collection and analysis techniques, was ignored. Consequently, future research adopting a qualitative research approach using interviews, observation, focus group, and others can be recommended.

## 5. Conclusions

This study of the impact of export promotion programs on export competitiveness and export performance of craft products aims to analyze the export promotion program, export competitiveness, and export performance using Structure Equation Modelling (SEM) to analyze the data from employees working in 46 exporting craft product companies in Thailand. The result indicated that export promotion programs, including financial and tax incentives, participation in trade fairs, export-related market research, export training, and export support and consulting, can influence the export competitiveness and export competitiveness including product quality, product differentiation, and promotion effectiveness can influence export performance in terms of international operation improvement, profitability, sales volume growth, and international opportunity. However, the study indicated that an export promotion program does not have a direct influence on export performance, but it has an indirect impact on export performance through export competitiveness. In other words, export competitiveness mediates the relationship between export promotion programs and export performance. Due to the study findings, managerial, theoretical, and political contributions can be remarked. For managerial contribution, the study suggests that entrepreneurs focus on product quality, product differentiation, and promotion effectiveness by producing unique products with high quality. As regards the theoretical contribution, the literature reviews related to the relationship among export promotion programs on export competitiveness and export performance of craft products can be firmed. As for the political contribution, the study reveals that the government should set the policy to promote export promotion programs, including direct promotion, knowledge and information sharing, consultancy, and investment support. In addition, the discussion of this study can contribute suggestions to marine transport, which is one of the critical international entry modes used by many businesses to expand businesses. Ultimately, shipping cost savings are related to export efficiency. However, the study has some limitations related to the variables, study area, and research methodology. Therefore, it is recommended that future research focuses more on potential variables that can influence the export competitiveness, extending the study area to other products as well as utilizing other research methods, such as a qualitative approach.

**Author Contributions:** Conceptualization, S.T.; writing—original draft preparation, S.T. and J.B.; writing review and editing, J.B. All authors have read and agreed to the published version of the manuscript.

**Funding:** This research received no external funding.

**Institutional Review Board Statement:** Certified by the Ethics Committee for Human Research of Silpakorn University. Code: COE 65.0215-043 Date of approval 15 February 2022.

**Informed Consent Statement:** Not applicable.

**Conflicts of Interest:** The authors declare no conflict of interest.

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
