# Peer review of "The Impact of Export Promotion Programs on Export Competitiveness and Export Performance of Craft Products"

_jmse, doi:10.3390/jmse10070892_

Round 1

Reviewer 1 Report

This paper is of sound quality on a subject deserving the Journal's attention. This study attempts to the impact of export promotion programs on export competitiveness and the performance of craft products. Employing Structure Equation Modelling, this study empirically analyse the casual relationship between variables. 1) the export promotion program has a significant positive relationship with export competitiveness, 2) A positive relationship between export competitiveness and export performance, and 3) The results indicate that export competitiveness fully mediates the relationship between export promotion programs and export performance. Overall, the paper is well written and well structured, therefore it is easy to follow and builds a clear conclusion from the data. Generally well written but requires some editing and revision.

 Yes. In literature review, this study well reviewed prior research, but there is no hypothesis development. To clearly present and highlight research objectives, I recommend putting hypotheses in literature review part.

 Yes, research design, data collection process and data analysis method are appropriate. Is there any justification of data collection period? for example... explain why the author captured the data from ~ to ~.

 Yes. The processes for data analysis are appropriate and the results of it are clearly described. However, this paper just described the results of data analysis. To improve the quality of this study, author(s) need to extract more clear implications in both theoretical and practical perspectives as a discussion of the results. Additional explanations are required to link the results of data analysis and conclusions.

 This study clearly presented the finding of this study, but research implication part is weak. Additional explanations incorporating theoretical and practical are required.

 The quality of communication is appropriate. Generally, well written but requires some editing and revision.

Author Response

Dear Editor in Chief and Reviewer,

Thank you for giving us the opportunity to submit a revised draft of the manuscript “Export promotion programs: The impact on export competitiveness and performance of craft products” for possible publication in the Journal of Maritime Science and Engineering. We appreciate the time and effort you and the reviewers dedicated to providing feedback on our manuscript. We are grateful for the insightful comments and valuable improvements to our paper. We have incorporated most of the suggestions made by the reviewers. The manuscript has been revised accordingly. Those changes and additional details are shown in track changes within the manuscript. Please see below for a point-by-point response to the reviewers’ comments.

Sincerely,

Authors

First of all, thank you so much for your delicated time in reviewing my article. Here is the revision version with using color to highlight the revision version.

Reviewer feedback

Author response/revision

Reviewer 1

This paper is of sound quality on a subject deserving the Journal's attention. This study attempts to the impact of export promotion programs on export competitiveness and the performance of craft products. Employing Structure Equation Modelling, this study empirically analyse the casual relationship between variables. 1) the export promotion program has a significant positive relationship with export competitiveness, 2) A positive relationship between export competitiveness and export performance, and 3) The results indicate that export competitiveness fully mediates the relationship between export promotion programs and export performance. Overall, the paper is well written and well structured, therefore it is easy to follow and builds a clear conclusion from the data. Generally well written but requires some editing and revision.

Thank you. The manuscript has been revised as per the comments provided.

 Yes. In literature review, this study well reviewed prior research, but there is no hypothesis development. To clearly present and highlight research objectives, I recommend putting hypotheses in literature review part.

I already put the hypothesis development along the literature review. The yellow color has been used to hightlight the revision version.

 Yes, research design, data collection process and data analysis method are appropriate. Is there any justification of data collection period? for example... explain why the author captured the data from ~ to ~.

The period of data collection has been added. The yellow color has been used to hightlight the revision version.

Yes. The processes for data analysis are appropriate and the results of it are clearly described. However, this paper just described the results of data analysis. To improve the quality of this study, author(s) need to extract more clear implications in both theoretical and practical perspectives as a discussion of the results. Additional explanations are required to link the results of data analysis and conclusions.

The implication hass been added into the discussion part. The yellow color has been used to hightlight the revision version.

This study clearly presented the finding of this study, but research implication part is weak. Additional explanations incorporating theoretical and practical are required.

The implication hass been added into the discussion part. The yellow color has been used to hightlight the revision version.

Reviewer 2 Report

The topic is interesting, and the author has done a great job in realizing the subject. However, there are few areas on the paper that is still lagging and should be addressed properly.

Abstract

1.     The authors should motivate the choice of variables  with theory and empirical backing on the subject

2.     Keywords should be revised to match key element of title

3.     Rewrite the title to be more catchy

4.     Introduction

1.     The objective of the paper presented need more clarifications to suit reader to understand the main idea of the paper especially for the study case is needed

2.     Literature review

 The literature is well written. However, there is need for more recent studies ranging from 2018-2022 to motivate the study properly. The entire study is too scanty and the related literature is not exhausted

The exportled growth in Malaysia: Does economic policy uncertainty and geopolitical risks matter?. Journal of Public Affairs22(1), e2361.

Methodology

1.     This section is generally well motivated, Kindly take note of the following minor additions

2.     More benefit of the various techniques utilized should be stated

3.     Check for cross-sectional dependency and add correlation text and VIF

4.     The authors should avoid much mathematical expressions or take some to appendix and make the study reader friendly for other practitioner other than academic with out compromise for study intend and quality.

Discussion

1.     The discussion is well written, but the authors should like their findings to the previous studies in the literature.

2.     There is need for professional proofreading or consult English native support

3.     Conclusion

1.     The sub-title should be conclusion and policy recommendation but not only conclusion

2.     The policy which is the engine of the study is weak and small. I therefore encourage the authors to elaborate more on the policy recommendations to policy makers for the investigated bloc

3.     The authors should add limitation of the study and future recommendation

Author Response

Dear Editor in Chief and Reviewer,

Thank you for giving us the opportunity to submit a revised draft of the manuscript “Export promotion programs: The impact on export competitiveness and performance of craft products” for possible publication in the Journal of Maritime Science and Engineering. We appreciate the time and effort you and the reviewers dedicated to providing feedback on our manuscript. We are grateful for the insightful comments and valuable improvements to our paper. We have incorporated most of the suggestions made by the reviewers. The manuscript has been revised accordingly. Those changes and additional details are shown in track changes within the manuscript. Please see below for a point-by-point response to the reviewers’ comments.

Sincerely,

Authors

First of all, I would like to say many thanks to your kind evaluation. Your feedback is so valuable for me which it can make my article more valuable. However, the revision of this article has been done in the article and explanation has been listed below.

Reviewer feedback

Author response/revision

Reviewer 2

The topic is interesting, and the author has done a great job in realizing the subject. However, there are few areas on the paper that is still lagging and should be addressed properly.

Abstract

1.    The authors should motivate the choice of variables  with theory and empirical backing on the subject

2.    Keywords should be revised to match key element of title

3.    Rewrite the title to be more catchy

Thank you. The manuscript has been revised as per the comments provided.

Here, the keyword and the title has been revised and used green to hightlight the edition version.

Introduction

1.      The objective of the paper presented need more clarifications to suit reader to understand the main idea of the paper especially for the study case is needed

Here, the objectives has been revised in order to be more ciryfied and used green to hightlight the edition version.

 Literature review

1.      The literature is well written. However, there is need for more recent studies ranging from 2018-2022 to motivate the study properly. The entire study is too scanty and the related literature is not exhausted

2.      The export‐led growth in Malaysia: Does economic policy uncertainty and geopolitical risks matter?. Journal of Public Affairs, 22(1), e2361.

This part added the article related to The export‐led growth in Malaysia: Does economic policy uncertainty and geopolitical risks matter?. Journal of Public Affairs, 22(1), e2361. in the literature review. Green color has been used to hightlight the edition version

Methodology

1.      This section is generally well motivated, Kindly take note of the following minor additions

2.      More benefit of the various techniques utilized should be stated

3.      Check for cross-sectional dependency and add correlation text and VIF

4.      The authors should avoid much mathematical expressions or take some to appendix and make the study reader friendly for other practitioner other than academic with out compromise for study intend and quality.

Here, correlation and VIF has been calculated and added into the acticle. Green color has been used to hightlight the edition version

Discussion

1.      The discussion is well written, but the authors should like their findings to the previous studies in the literature.

2.      There is need for professional proofreading or consult English native support

Green color has been used to hightlight the edition version

Conclusion

1.      The sub-title should be conclusion and policy recommendation but not only conclusion

2.      The policy which is the engine of the study is weak and small. I therefore encourage the authors to elaborate more on the policy recommendations to policy makers for the investigated bloc

3.      The authors should add limitation of the study and future recommendation

The sub-title, contribution and limitation hass been added into the conclusion part. Green color has been used to hightlight the edition version

Reviewer 3 Report

Thanks for the opportunity to review the article. the topic is interesting, but there are places to fix:

1. The introduction should highlight the issues and the main purpose of the article.

2. Statistics should be provided in Part 2 of the work

3. The survey methodology should specify the sample size from which the 400 respondents were estimated.

4. There is a lot of citation in the discussion section. The authors of the article lack a critical approach to the analysis of the results.

5. The conclusions do not indicate the directions of further research.

Author Response

Dear Editor in Chief and Reviewer,

Thank you for giving us the opportunity to submit a revised draft of the manuscript “Export promotion programs: The impact on export competitiveness and performance of craft products” for possible publication in the Journal of Maritime Science and Engineering. We appreciate the time and effort you and the reviewers dedicated to providing feedback on our manuscript. We are grateful for the insightful comments and valuable improvements to our paper. We have incorporated most of the suggestions made by the reviewers. The manuscript has been revised accordingly. Those changes and additional details are shown in track changes within the manuscript. Please see below for a point-by-point response to the reviewers’ comments.

Sincerely,

Authors

First of all, I am really appreciated your comments and evalution. Your feedback is so valuable and can lead this article more appropriate. From your feedback, I have been editing as follows.

Reviewer feedback

Author response/revision

Reviewer 3

Thanks for the opportunity to review the article. the topic is interesting, but there are places to fix:

1.      The introduction should highlight the issues and the main purpose of the article.

Thank you. The manuscript has been revised as per the comments provided.

The introduction is edited and blue has been used to hightlight the edition version.

2.      Statistics should be provided in Part 2 of the work

I inserted more statistics and used blue to hightlight the edition version.

3.      The survey methodology should specify the sample size from which the 400 respondents were estimated.

I use Cochran’s method to derive the sample size. Blue color has been used to hightlight the edition version.

4.      There is a lot of citation in the discussion section. The authors of the article lack a critical approach to the analysis of the results.

I add some sentences and used blue to hightlight the edition version.

5.      The conclusions do not indicate the directions of further research.

I add some sentences and used blue to hightlight the edition version.

Round 2

Reviewer 3 Report

You write "The population in this study were craft product companies, but the numbers of the population were unknown; therefore, the study samples were drawn by using Cochran's method [35].", But given that "The Cochran formula allows you to calculate an ideal sample size given the desired level of precision, desired confidence level, and the estimated proportion of the attribute present in the population. Cochran's formula is considered particularly appropriate in situations with large populations. " - the wording of the sentence in question should be further clarified or adjusted. Because this method is applied to knowing the exact numbers.

Author Response

Dear Editor in Chief and Reviewer,

Thank you for giving us the opportunity to submit a revised draft of the manuscript titled, "Export promotion programs: The impact on export competitiveness and performance of craft products" for publication in the Journal of Maritime Science and Engineering. We appreciate the time and effort you and the reviewers have dedicated to provide feedbacks on our manuscript. We are grateful for the insightful comments and suggestions to improve our paper. The manuscript has been revised according to the comments and suggestions. Those changes and additional details are shown in track changes within the manuscript. Please see below for our responses/revisions to the reviewers' comments.

Sincerely,

Authors

Reviewer feedback

Author response/revision

Reviewer 3

You write "The population in this study were craft product companies, but the numbers of the population were unknown; therefore, the study samples were drawn by using Cochran's method [35].", But given that "The Cochran formula allows you to calculate an ideal sample size given the desired level of precision, desired confidence level, and the estimated proportion of the attribute present in the population. Cochran's formula is considered particularly appropriate in situations with large populations. " - the wording of the sentence in question should be further clarified or adjusted. Because this method is applied to knowing the exact numbers.

Thank you. The manuscript has been revised as the comments provided.

According to your suggestion, we have added an explanation sample size calculation method for Cochran's method. An explanation of the method is provided.
